# Hierarchical Enhancement Framework for Aspect-based Argument Mining

**Yujie Fu[1], Yang Li[2], Suge Wang[1,3]\*, Xiaoli Li[4], Deyu Li[1,3], Jian Liao[1], JianXing Zheng[1]**

[1] School of Computer and Information Technology, Shanxi University, China
[2] School of Finance, Shanxi University of Finance and Economics, China
[3] Key Laboratory of Computational Intelligence and Chinese Information Processing
of Ministry of Education, Shanxi University, China
[4] Institute for Infocomm Research, A*STAR, Singapore

## Abstract

Aspect-Based Argument Mining (ABAM) is a critical task in computational argumentation. Existing methods have primarily treated ABAM as a nested named entity recognition task, overlooking the need for tailored strategies to effectively address the specific challenges of ABAM tasks. To this end, we propose a layer-based **H**ierarchical **E**nhancement **F**ramework (HEF) for ABAM, and introduce three novel components: the Semantic and Syntactic Fusion (SSF) component, the Batch-level Heterogeneous Graph Attention Network (BHGAT) component, and the Span Mask Interactive Attention (SMIA) component. These components serve the purposes of optimizing underlying representations, detecting argument unit stances, and constraining aspect term recognition boundaries, respectively. By incorporating these components, our framework enables better handling of the challenges and improves the performance and accuracy in argument unit and aspect term recognition. Experiments on multiple datasets and various tasks verify the effectiveness of the proposed framework and components. [1]

## 1 Introduction

Argument mining, a critical task within computational argumentation, has gained considerable attention, evident from available datasets (Stab et al., 2018; Trautmann et al., 2020), emerging tasks (Wachsmuth et al., 2017; Al-Khatib et al., 2020), and machine learning models associated to this domain (Kuribayashi et al., 2019; Chakrabarty et al., 2019). As aspect-level sentiment analysis tasks have flourished, Aspect-Based Argument Mining (ABAM) has recognized the need to decompose *argument units* into smaller attributes and define *aspect terms* as components with specific meanings in arguments (Trautmann, 2020).

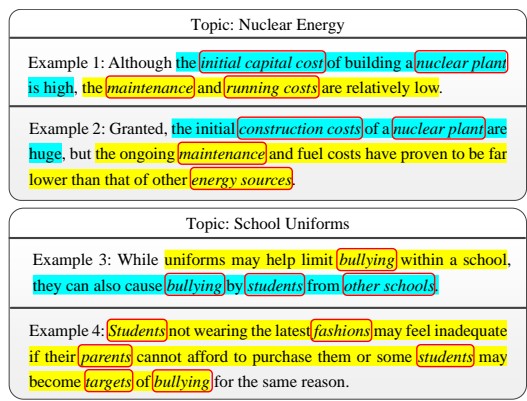

Figure 1: Example annotation of the argument units, the corresponding stances (yellow: supporting/pro; blue: opposing/con) and the aspect term (italics framed in red) for the topics *nuclear energy* and *school uniforms*.

Previous works have attempted to combine aspects and arguments, but they often lack a clear definition of the relevant task. For instance, Fujii and Ishikawa (2006) primarily focuses on summarizing viewpoints and defining auguring points. Similarly, Misra et al. (2015) further groups the arguments under discussion into aspects of the argument. Furthermore, Gemechu and Reed (2019) considers aspects in argument relations as part of four functional components. Only recently, a study by Trautmann (2020) specifically addresses aspect term extraction and emphasizes the definition, introducing the concept of **Aspect-Based Argument Mining** (ABAM). *The main objective of ABAM is to identify the **argument units** that support corresponding stances under a controversial topic, along with the **aspect terms** mentioned within these argument units.* In this context, an *argument unit* is typically defined as a short text or span, providing evidence or reasoning about the topic, supporting or opposing it (Stab et al., 2018). On the other hand, an *aspect term* is defined as the crucial facet/part the argument unit is trying to address, representing a core aspect of the argument (Trautmann, 2020).

---

\*Corresponding author. Email: wsg@sxu.edu.cn.

[1]Codes available at https://github.com/sxu-fyj/ABAM.

Figure 1 illustrates four examples within the topic of *nuclear energy* and *school uniforms*. For the first set of examples (example 1 and example 2), argument unit (yellow or blue) opinion are expressed around aspect terms (italics framed in red), such as *cost*, *nuclear plant*, and *maintenance*. Similarly, the second set of examples (example 3 and example 4) revolves around several aspect terms such as *students*, and *bullying*. This targeted approach enables a more precise analysis by zooming in on specific aspects that are essential to the **argument unit**. Moreover, these aspect terms enable the comparison of support or opposing opinions at the aspect level, thereby facilitating the acquisition of more nuanced and fine-grained conclusions.

ABAM, as defined by Trautmann (2020), is treated as a Nested Named Entity Recognition (NNER) task which presents **three key challenges**: 1) How to *construct a robust underlying representation to effectively encode contextual information*? In the realm of Natural Language Processing (NLP), the significance of a robust underlying representation serves as a cornerstone for achieving excellent model performance. 2) How to *mine the correlation between opinion expressions corresponding to different stances under the same topic*? Since different users may give different expressions of viewpoints on same stances within a given topic. As shown in Figure 1, authors express different opinions around similar aspect terms under the same topic. Exploring the relationship between these opinion expressions can greatly assist in determining the corresponding stances of different argument units accurately. 3) How to *leverage task-specific features to improve the extraction of argument units and aspect terms*? By investigating the unique task characteristics and data properties of the ABAM task, we aim to enhance the model's performance significantly.

Overall, we propose a novel **H**ierarchical **E**nhancement **F**ramework (HEF), consisting of four modules: basic module, argument unit enhancement module, aspect term enhancement module, and decision module. With regard to the three challenges above, the paper presents three key components accordingly. In the basic module, we propose the Semantic and Syntactic Fusion (SSF) component to fine-tune the representation of the pre-trained language model. This fine-tuning helps us complete the *initial recognition stage of argument units and aspect terms*. Next, in the argument unit

enhancement module, we leverage the argument unit boundary information provided by the basic module. By integrating a span-based method and utilizing the proposed Batch-Level Heterogeneous Graph Attention Network (BHGAT) component, we are able to judge the stance of the argument unit, thereby *refining the categorization of the initially recognized argument units*. Moving on to the aspect term enhancement module, we introduce the Span Mask Interactive Attention (SMIA) component. By incorporating span masks and interactive guidance through attention mechanisms, we can *better capture and identify aspect terms* within the specified boundaries. Finally, in the decision module, we combine the initial recognition results with the enhancement results to produce the final output. Our contribution can be summarized as follows:

- We propose a general SSF component to enhancing underlying representations, which can simultaneously capture both semantic and syntactic information.

- We propose a novel framework BHGAT, which can seamlessly integrates the strengths of two distinct types of Graph Neural Networks (GNNs) within a batch-level setting.

- We propose a task-special SMIA component, which used to effectively constrain the recognition range of aspect terms.

- Experiments on multiple datasets and various tasks verify the effectiveness of the proposed framework and components.

## 2 Related Work

The objective of Misra et al. (2015) is to identify specific arguments and counter-arguments in social media texts, categorize them into different aspects, and utilize this aspect information to generate argument summaries. Similarly, Misra et al. (2016) focus on inducing and identifying argument aspects across multiple conversations, ranking the extracted arguments based on their similarity, and generating corresponding summaries. However, these earlier works have been limited to a few specific topics. In recent research, the approach has been extended to cover a broader range of 28 topics, introducing a novel corpus for aspect-based argument clustering (Reimers et al., 2019). Furthermore, Gemechu and Reed (2019) decompose propositions into four functional components: aspects, target concepts, and opinions on aspects and target concepts. By leveraging the relationships

among these components, they infer argument relations and gain a deeper understanding of the argument structure. In a different study, Bar-Haim et al. (2020) focus on summarizing the arguments, supporting each side of a debate, mapping them to a concise list of key points, which are similar to the aspect terms highlighted earlier. Lastly, Trautmann (2020) redefines the aspect-based argument mining task based on clause-level argument unit recognition and classification in heterogeneous document collections (Trautmann et al., 2020).

## 3 Framework

This paper proposes a HEF for aspect-based argument mining task, which consists of four modules: basic module, argument unit enhancement module, aspect term enhancement module and decision module. The architecture of the HEF is visually depicted in figure 2.

### 3.1 Task Definition

Following Trautmann (2020), we formulate the ABAM task as a nested named entity recognition task with a two-level nested structure. Given an input sentence of $n$ tokens $W^{text} = [w_1^{text}, w_2^{text}, ..., w_n^{text}]$, the topic is $W^{topic} = [w_1^{topic}, ..., w_m^{topic}]$, the target argument unit (AU) label sequence is $Y^{AU} = [y_1^{AU}, y_2^{AU}, ..., y_n^{AU}]$ and the target aspect term (AT) label sequence is $Y^{AT} = [y_1^{AT}, y_2^{AT}, ..., y_n^{AT}]$, where $y^{AU} \in \{B_{con}, I_{con}, E_{con}, B_{pro}, I_{pro}, E_{pro}, O\}$ and $y^{AT} \in \{B_{asp}, I_{asp}, E_{asp}, O\}$.

### 3.2 Basic Module

The sentence and topic are concatenated as the input to BERT: [CLS], $w_1^{text}, ..., w_n^{text}$, [SEP], $w_1^{topic}, ..., w_m^{topic}$, [SEP], where [CLS] and [SEP] are special tokens. The contextualized representations of each token $X = [x_1^w, x_2^w, ..., x_n^w]$ can be given as:

$$x_t^w = \text{BERT}(w_t^{text}) \tag{1}$$

Note we also incorporate orthographic and morphological features of words by combining character representations (Xu et al., 2021). The characters representation with in $w_i^{text}$ as $w_i^{char}$. Then we use LSTM to learn the final hidden state $x_t^c$ as the character representation of $w_i^{text}$:

$$x_t^c = \text{LSTM}(w_t^{char}) \tag{2}$$

The final token representation is obtained as follows:

$$x_t = [x_t^w; x_t^c; x_t^p] \tag{3}$$

where $[;]$ denotes concatenation, and $x_t^p$ is the part-of-speech tagging of $w_t^{text}$.

**Encoder.** The LSTM is widely used for capturing sequential information in either the forward or backward direction. However, it faces challenges when dealing with excessively long sentences, as it may struggle to retain long-distance dependencies between words. To address this limitation and exploit syntactic information, we propose the Semantic and Syntactic Fusion (SSF) component by sentence-level GNNs, aiming to bridge the gap between distant words by effectively encoding both sequential semantic information and spatial syntactic information.

The input of SSF: previous cell state $c_{t-1}$, previous hidden state $h_{t-1}$, current cell input $x_t$, and an additional graph-encoded representation $g_t$, where $c_0$ and $h_0$ are initialized to zero vectors, $g_t^{(l)}$ is a graph-encoded representation generated using Graph Attention Network (GAT), which are capable of bringing in structured information through graph structure (Hamilton et al., 2017).

The hidden state $h_t$ of SSF are computed as follows:

$$f_t = \sigma(W^{(f)}x_t + U^{(f)}h_{t-1} + b^{(f)}) \tag{4}$$

$$i_t = \sigma(W^{(i)}x_t + U^{(i)}h_{t-1} + b^{(i)}) \tag{5}$$

$$o_t = \sigma(W^{(o)}x_t + U^{(o)}h_{t-1} + b^{(o)}) \tag{6}$$

$$\tilde{c}_t = \tanh(W^{(c)}x_t + U^{(c)}h_{t-1} + b^{(c)}) \tag{7}$$

$$c_t = f_t * c_{t-1} + i_t * \tilde{c}_t \tag{8}$$

$$m_t = \sigma(W^{(m)}x_t + U^{(m)}h_{t-1} + Q^{(m)}g_t^{(l)} + b^{(m)}) \tag{9}$$

$$s_t = \tanh(W^{(s)}x_t + U^{(s)}h_{t-1} + Q^{(s)}g_t^{(l)} + b^{(s)}) \tag{10}$$

$$h_t = o_t * \tanh(c_t) + m_t * \tanh(s_t) \tag{11}$$

where, $f_t$, $i_t$, $o_t$, $c_t$ are equivalent to the forget gate, input gate, output gate, and cell unit in the traditional LSTM respectively, $m_t$ and $s_t$ are used to control the information flow of $g_t^{(l)}$. Finally $h_t$ is the output of SSF.

Star-Transformer (Guo et al., 2019) can measure the position information more explicitly and is more sensitive to the order of the input sequence. Building upon this insight, we use the output of SSF component as the input of Star-Transformer to re-encode the context to complete the encoder part.

$$h_t^{star} = \text{Star-Transformer}(h_t) \tag{12}$$

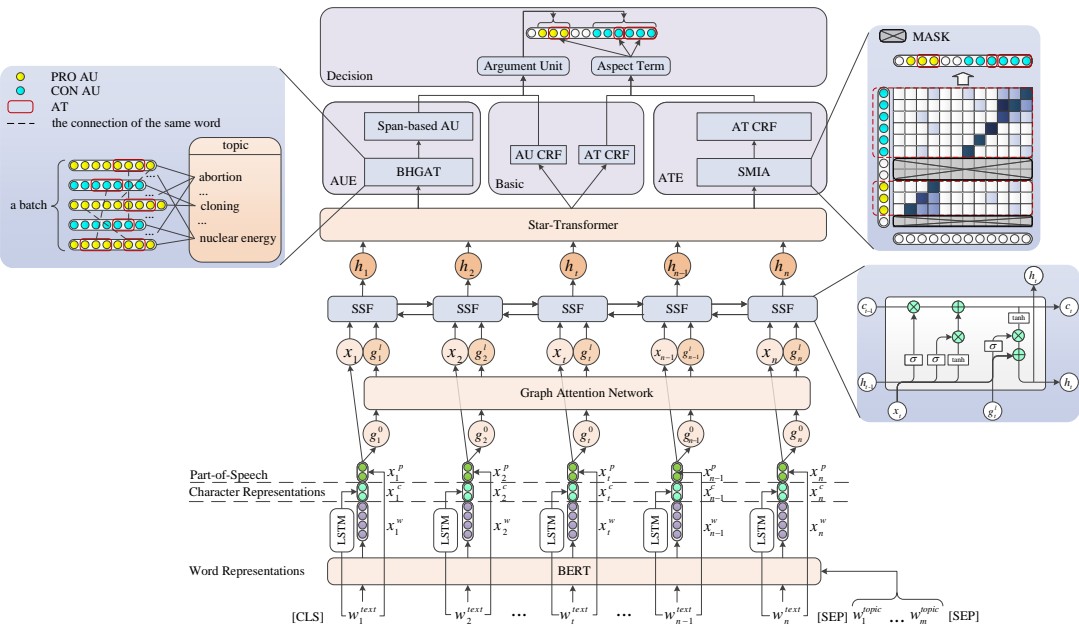

Figure 2: The overall architecture of HEF for ABAM.

**Decoder.** CRF has been widely used in NER task (Xu et al., 2021; Li et al., 2021). For an input sentence, the probability scores $z_t^{AU}$ and $z_t^{AT}$ for all tokens $x_i \in X$ over the argument unit tags and aspect term tags are calculated by CRF decoder:

$$z_t^{AU} = p(y_t^{AU}|h_t^{star}) = CRF(W^{AU}h_t^{star} + b^{AU}) \quad (13)$$

$$z_t^{AT} = p(y_t^{AT}|h_t^{star}) = CRF(W^{AT}h_t^{star} + b^{AT}) \quad (14)$$

### 3.3 Argument Unit Enhancement Module

Motivated by span-based methods, we utilize argument unit labels $z_t^{AU}$ that predicted by the basic module to obtain boundary information of argument unit spans, and re-evaluating the stance of each span, thereby correcting the $z_t^{AU}$ labels, which is argument unit enhancement module (AUE). We observe that different users often express similar opinions, when discussing similar aspect terms. Learning these similar opinion expressions can assist in distinguishing the corresponding stances of argument units. Furthermore, pre-trained language models are widely adopted due to their ability to generate robust contextual representations. However, different contexts can yield different representations for the same word. Exploring the correlation between different context representations of the same word can aid in optimizing the underlying representations. To this end, we propose the Batch-level Heterogeneous Graph Attention Network (BHGAT) component. BHGAT combines

the strengths of sentence-level GNNs (Zhang et al., 2019; Wang et al., 2020) and corpus-level GNNs (Wang et al., 2019; Yao et al., 2019; Linmei et al., 2019). While utilizing the deep contextual word representations generated by pre-trained language models, BHGAT facilitates communication of opinion expressions among different samples and establishes correlations between different representations of the same word. This enables the optimization of the representations of various heterogeneous nodes within the graph. Constructing a Graph Neural Network involves defining an initial representation for each node, an adjacency matrix, and a node update method.

**Node initialization.** In our proposed BHGAT, we distinguish between two types of nodes: argument unit nodes $h_i^{au}$ and word nodes $h_t^{star}$. The specific operation is as follows:

$$h_i^{au} = [h_{start_i}^{star}; h_{end_i}^{star}; au_i(topic)] \quad (15)$$

where $h_{start_i}^{star}$ and $h_{end_i}^{star}$ are the starting and ending word representation of $i$-th argument unit $au_i$, and $au_i(topic)$ is the topic of $au_i$.

$$HG^{(0)} = [h_1^{au}, ..., h_{n_{au}}^{au}, h_1^{star}, ..., h_{n_w}^{star}] \quad (16)$$

where $HG^{(0)}$ is initial representation of nodes in BHGAT, $n_{au}$ is the number of argument units, and $n_w$ is the number of words.

**Adjacency matrix**. The adjacency matrix A of BHGAT includes 4 parts: [$au$, $au$] part, [$au$, $word$]

part, [*word*, *au*] part, [*word*, *word*] part.

$$A = \begin{bmatrix} [au, au] & [au, word] \\ [word, au] & [word, word] \end{bmatrix} \quad (17)$$

The $(au_i, au_j)$ captures the relationships between argument units within a batch, facilitating communication and understanding among argument units that share the same topic, and allowing us to learn similar opinion expressions:

$$(au_i, au_j) = \begin{cases} 1, & if\ au_i(topic) = au_j(topic) \\ 0, & otherwise \end{cases}$$

$$(18)$$

The $(au_i, word_j)$ represents the association between argument units and words, using the attention mechanism between nodes to complete the update of argument unit nodes and word nodes, which is:

$$(au_i, word_j) = \begin{cases} 1, & if\ word_j\ in\ au_i \\ 0, & otherwise \end{cases} \quad (19)$$

Similarly, the $(word_i, au_j)$ is denoted as:

$$(word_i, au_j) = \begin{cases} 1, & if\ word_i\ in\ au_j \\ 0, & otherwise \end{cases} \quad (20)$$

The $(word_i, word_j)$ focuses on node representations of the same word in different contexts. This part facilitates the information interaction of word nodes between different argument units, which is conducive to optimize the dynamic representations of the underlying words.

$$(word_i, word_j) = \begin{cases} 1, & if\ word_i = word_j\ \text{in different AU} \\ 0, & otherwise \end{cases}$$

$$(21)$$

Furthermore, the diagonal values of $A$ are all ones.

**Node update.** We adopt the method of information aggregation to complete node updates, similar to GAT (Veličković et al., 2018). The specific operation is as follows:

$$\alpha_{ij} = \frac{exp(\text{LeakyReLU}(a^T[Whg_i\|Whg_j]))}{\sum_{k\in N_i} exp(\text{LeakyReLU}(a^T[Whg_i\|Whg_j]))} \quad (22)$$

$$hg_i^{(l+1)} = \sigma\left(\frac{1}{K}\sum_{k=1}^{K}\sum_{j\in N_i}\alpha_{ij}^k W^k hg_j^{(l)}\right) \quad (23)$$

where $hg_i^{(l)}$ is the representation of nodes for $l$-th layer.

Finally, we perform stance classification on the representations of argument unit nodes. The probability $p_{au_i}$ of different stance belonging to $au_i$ is calculated as follows:

$$p_{au_i} = softmax(W^{hg}hg_i^{(l)} + b^{hg}), i \in \{1,...,n_{au}\} \quad (24)$$

where $p_{au_i} = [p_{au_i}^{con}, p_{au_i}^{pro}, p_{au_i}^{non}]$ is the stance probability distribution of the argument unit $au_i$.

Through BHGAT, we first obtain the stance classes of the corresponding argument units. Then, we map the obtained stance classes to the enhanced argument unit recognition space, resulting in the vector $z_t^{AUE}$. $z_t^{AUE}$ can be divided into two parts: the boundary part and the stance part. In the mapping process, first, according to the boundary of the argument unit label $z_t^{AU}$, it is judged whether the boundary of $z_t^{AUE}$ corresponding to the $token_t$ is B-*, I-*, E-* or O. Then according to $p_{s_i}^{con}, p_{s_i}^{pro}, p_{s_i}^{non}$, we determine the stance part in $z_t^{AUE}$ (*-con, *-pro, O) as follows:

$$z_t^{AUE} = \begin{cases} [p_{au_i}^{con}, 0, 0, p_{au_i}^{pro}, 0, 0, p_{au_i}^{non}], & t = s_{au_i} \\ [0, p_{au_i}^{con}, 0, 0, p_{au_i}^{pro}, 0, p_{au_i}^{non}], & s_{au_i} < t < e_{au_i} \\ [0, 0, p_{au_i}^{con}, 0, 0, p_{au_i}^{pro}, p_{au_i}^{non}], & t = e_{au_i} \\ [0, 0, 0, 0, 0, 0, 1], & otherwise \end{cases}$$

$$(25)$$

where $s_{au_i}$ is the start position of argument unit $au_i$ and $e_{au_i}$ is the end position.

### 3.4 Aspect Term Enhancement Module

To enhance the label sequence results $z_t^{AT}$ of initially identified aspect terms, we introduce the Aspect Term Enhancement (ATE) module. Since an aspect term is specific to a corresponding argument unit, it is essential to establish a component that constrains the recognition range of aspect terms within the text. Building upon this concept, we propose the Span Mask Interactive Attention (SMIA) component, which ensures that the attention mechanism focuses on the relevant argument unit spans while effectively disregarding irrelevant text. The overall process can be formulated as:

$$H^{mask} = Att(H^{star}W^Q, H^{star}W^K, H^{star}W^V) \quad (26)$$

$$Att(Q, K, V) = softmax(MASK + \frac{QK^T}{\sqrt{d_k}}) \cdot V \quad (27)$$

$$mask_t = \begin{cases} -\infty, & y_t^{AU} = O \\ 0, & y_t^{AU} \neq O \end{cases} \quad (28)$$

Once we obtain the new context representation, we proceed to feed it into the decoder model, which

is responsible for generating the aspect term enhanced label sequence $z_t^{ATE}$.

$$z_t^{ATE} = p(y_t^{AT}|h_t^{mask}) = CRF(W^{ATE}h_t^{mask} + b^{ATE}) \quad (29)$$

## 3.5 Decision Module

Through the AUE and ATE module, we can obtain the enhanced argument unit label probability $z_t^{AUE}$ and the enhanced aspect term label probability $z_t^{ATE}$. Finally, we fuse the probabilities in the two label spaces (initial, enhanced) as the final output.

$$\tilde{z}_t^{AT} = z_t^{ATE} + z_t^{AT} \quad (30)$$

$$\tilde{z}_t^{AU} = z_t^{AUE} + z_t^{AU} \quad (31)$$

## 3.6 Objective Function

The first part aims to minimize two negative log-probability of the correct sequence of labels in basic module.

$$L_{NER} = -\sum_{t=1}^{T}\log(p(y_t^{AT}|z_t^{AT})) - \sum_{t=1}^{T}log(p(y_t^{AU}|z_t^{AU})) \quad (32)$$

where $z_t^{AU}$ and $z_t^{AT}$ represent the predicted sequence, $y_t^{AU}$ and $y_t^{AT}$ represent the correct sequence.

The second part loss is the cross-entropy loss for stance classification of argument unit span in AUE module, denoted as:

$$L_{AUE} = -\sum_{i=1}^{n}\sum_{j=1}^{m}y_{au_i}^{stance_j}\log(p_{au_i}^{stance_j}) \quad (33)$$

where $n$ is the number of argument units, and $m$ is the number of stance classes.

Similar to the first part, the third part uses the negative log-likelihood loss in ATE module.

$$L_{ATE} = -\sum_{t=1}^{T}\log(p(y_t^{AT}|z_t^{ATE})) \quad (34)$$

Finally, the fourth part also aim to minimize negative log-likelihood for enhanced label probability distribution.

$$L_E = -\sum_{t=1}^{T}\log(p(y_t^{AT}|\tilde{z}_t^{AT})) - \sum_{t=1}^{T}log(p(y_t^{AU}|\tilde{z}_t^{AU})) \quad (35)$$

The final loss function is defined as follows:

$$L = L_{NER} + L_{AUE} + L_{ATE} + L_E \quad (36)$$

## 4 Experiments

To evaluate the effectiveness of HEF framework and the corresponding components, we conducted experiments on four datasets.

### 4.1 Datasets

**ABAM**[2]. We employ the latest ABAM dataset, which was released in 2020 and comprises 8 topics (Trautmann, 2020). The statistics are presented in the table 1. We followed the inner dataset split (2268 / 307 / 636 for train / dev / test) defined in the ABAM corpus (Trautmann, 2020).

Table 1: The proportion of ABAM.

| topic | #sentence | #segment | #aspect(total) | #aspect(unique) |
|---|---|---|---|---|
| abortion | 415 | 435 | 910 | 484 |
| cloning | 343 | 365 | 843 | 492 |
| marijuana legalization | 626 | 676 | 1889 | 887 |
| minimum wage | 624 | 689 | 1981 | 745 |
| nuclear energy | 615 | 671 | 1992 | 980 |
| death penalty | 588 | 637 | 1325 | 545 |
| gun control | 480 | 519 | 1081 | 429 |
| school uniforms | 705 | 800 | 2019 | 923 |
| total | 4396 | 4792 | 12040 | 4525 |

**AURC-8**[3]. The argument unit recognition and classification (AURC) dataset published in 2020, consists of 8 topics (Trautmann et al., 2020). The statistics are shown in the table 2. We used the inner dataset split (4000 / 800 / 2000 for train / dev / test) given by Trautmann et al. (2020).

Table 2: The proportion of AURC-8.

| topic | #number | #arg-sent | #non-arg | #arg-unit |
|---|---|---|---|---|
| abortion | 1000 | 424 | 576 | 458 |
| cloning | 1000 | 353 | 647 | 380 |
| marijuana legalization | 1000 | 630 | 370 | 689 |
| minimum wage | 1000 | 630 | 370 | 703 |
| nuclear energy | 1000 | 623 | 377 | 684 |
| death penalty | 1000 | 598 | 402 | 651 |
| gun control | 1000 | 529 | 471 | 587 |
| school uniforms | 1000 | 713 | 287 | 821 |
| total | 8000 | 4500 | 3500 | 4973 |

**SemEval-2016 Task 6A**[4]. The dataset has been divided into training and test set for each of the five claims. Each sample can be classified three categories: against, none and favor.

**ABAM argument segment**[5]. The dataset is a collection of argument units in ABAM. Each argument unit can be classified into two categories: PRO and CON.

Table 3 shows the distribution of SemEval-2016 Task 6A and ABAM argument segment.

[2]https://github.com/trtm/ABAM
[3]https://github.com/trtm/AURC
[4]https://github.com/sxu-fyj/stance
[5]https://github.com/trtm/ABAM

Table 3: The proportion of SemEval-2016 Task 6A and ABAM argument segment.

| | Train | Dev | Test | total |
|---|---|---|---|---|
| SemEval-2016 Task 6A | 2914 | - | 1249 | 4163 |
| ABAM argument segment | 2447 | 333 | 693 | 3473 |

## 4.2 The experimental setup

**Evaluation Metrics:** For different tasks, we provide specific evaluation metrics to assess the performance. For **aspect-based argument mining task**, we conduct model evaluation using the ABAM dataset at the segment-level and token-level. In *segment-level evaluation*, we consider a prediction as correct only if the model correctly identifies the boundary and category of an entity. We use exact matching F1[6] to measure the accuracy. At the token-level, we proposed two evaluation methods: Token-Nested evaluation and Token-Flat evaluation. In the *Token-Nested evaluation*, we extend the stance labels by incorporating aspect information, resulting in six possible combinations: [NON, O], [NON, ASP], [PRO, O], [PRO, ASP], [CON, O], and [CON, ASP]. We report both Macro-F1 and Micro-F1 scores for this evaluation. In *Token-Flat evaluation*, we concatenate the sequence labels of aspect term recognition and argument unit recognition to generate a label sequence twice the sentence length, containing four label categories: [ASP, PRO, CON, O]. We report both Macro-F1 and Micro-F1 scores for this evaluation. For the **argument unit recognition and classification** task, we employ the segment-level evaluation metric on the AURC-8 dataset. The Macro-F1, Micro-F1 and separate F1 scores for each category are provided. Finally, for the **stance detection task**, we report the Macro-F1 score, Micro-F1 score, and F1 scores on the SemEval-2016 Task 6A and ABAM argument segment datasets.

**Compared Models:** We compare the HEF framework with the following state-of-the-art methods:

- **CNN-NER** (Yan et al., 2022) utilizes a convolutional neural network to capture the interactions among neighboring entity spans.

- **W²NER** (Li et al., 2022) introduces a novel approach to tackle named NER by framing it as a word-word relation classification problem.

---

[6]https://github.com/chakki-works/seqeval

- **Span, BPE, Word** (Yan et al., 2021) present a new formulation of the NER task as an entity-span sequence generation problem.

## 4.3 Experimental results and analysis

### 4.3.1 Comparison results of different methods for ABAM

We show the performance comparison of different methods in table 4. All comparison methods use the code provided by their respective original papers, and undergo testing and evaluation on the ABAM dataset.

Based on the results presented in Table 4, our proposed method demonstrates superior performance compared to all the existing methods, both in segment-level and token-level evaluation metrics. This improvement can be attributed to the inclusion of two key modules: Argument Unit Enhancement (AUE) and Aspect Term Enhancement (ATE). Specifically, our method shows substantial improvements in MicF1, MacF1, Token-Flat, and Token-Nested evaluation metrics, with gains of at least 0.0767, 0.1274, 0.0647, and 0.0745, respectively, compared to the other comparison methods. The ATE module effectively constrains the recognition range of aspect terms, leading to a significant improvement in the F1 score for aspect term recognition (ASP column).

### 4.3.2 Ablation experiments for ABAM

To evaluate the individual impact of each functional module or component within the HEF framework on model performance, we conduct a series of ablation experiments. The results of these experiments are presented in Table 5. -w/o topic, -w/o SSF, -w/o Star-Transformer, -w/o AUE and -w/o ATE represents our model after removing the topic in BERT, SSF, Star-Transformer, AUE module, and ATE module, respectively.

The experimental results in the table above clearly demonstrate that the removal of different modules or components has a significant impact on the performance of the HEF framework. In particular, the absence of the AUE module has a substantial negative effect on overall performance. The utilization of BHGAT to re-judge the category of argument unit spans has proven to be an effective strategy for correcting samples in the basic module where the boundary is correctly identified but the category judgment is incorrect. Moreover, the inclusion of the topic information in BERT

Table 4: Performance comparison of different methods in ABAM.

| | Segment-Level | | | | | Token-Level | |
|---|---|---|---|---|---|---|---|
| | ASP | CON | PRO | MicF1 | MacF1 | Token-Flat | Token-Nested |
| CNN-NER(Yan et al., 2022) | 0.7480 | 0.4535 | 0.4491 | 0.6707 | 0.5502 | - | - |
| W$^2$NER(Li et al., 2022) | 0.7495 | 0.4618 | 0.4495 | 0.6727 | 0.5536 | 0.7587 | 0.5347 |
| Word(Yan et al., 2021) | 0.7191 | 0.4423 | 0.4650 | 0.6474 | 0.5421 | 0.7656 | 0.5438 |
| Span(Yan et al., 2021) | 0.7175 | 0.4657 | 0.4702 | 0.6516 | 0.5511 | 0.7680 | 0.5459 |
| BPE(Yan et al., 2021) | 0.7090 | 0.3695 | 0.3560 | 0.6265 | 0.4782 | 0.6676 | 0.4344 |
| HEF | **0.7969** | **0.6283** | **0.6176** | **0.7494** | **0.6810** | **0.8327** | **0.6204** |

Table 5: Performance comparison of results without different module or components.

| | Segment-Level | | | | | Token-Level | |
|---|---|---|---|---|---|---|---|
| | ASP | CON | PRO | MicF1 | MacF1 | Token-Flat | Token-Nested |
| HEF | **0.7969** | **0.6283** | **0.6176** | **0.7494** | **0.6810** | **0.8327** | **0.6204** |
| -w/o topic | 0.7921 | 0.6073 | 0.6083 | 0.7416 | 0.6693 | 0.8238 | 0.6096 |
| -w/o SSF | 0.7911 | 0.4460 | 0.4646 | 0.6963 | 0.5672 | 0.8008 | 0.5735 |
| -w/o Star-Transformer | 0.7851 | 0.4715 | 0.4823 | 0.6980 | 0.5797 | 0.7845 | 0.5574 |
| -w/o AUE | 0.7367 | 0.4276 | 0.4249 | 0.6525 | 0.5298 | 0.7716 | 0.5568 |
| -w/o ATE | 0.7399 | 0.4620 | 0.4843 | 0.6667 | 0.5621 | 0.7700 | 0.5462 |

contributes significantly to the framework's performance. The SSF and Star-Transformer components also play crucial roles in generating high-quality underlying representations. The absence of these components has a detrimental impact on the model's performance. Lastly, due to aspect terms only exist within the argument unit spans. The addition of the ATE module improves the accuracy of aspect term range extraction.

### 4.3.3 Effectiveness of SSF component

To evaluate the effectiveness of SSF component, we conducted experiments on two sequence labeling datasets, AURC-8 and ABAM. For the sequence labeling task, we use LSTM and SSF as the encoder and CRF as the decoder. The experimental results are presented in Table 6.

Based on the experimental results in Table 6, we observe the scores on the ABAM dataset are significantly higher compared to those in the AURC-8 dataset. This discrepancy can be attributed to the inherent dissimilarities between the two datasets. In the AURC-8 dataset, argument units may not exist in every sample, while the ABAM dataset ensures the presence of at least one argument unit in each sample. By integrating spatial syntactic information with the LSTM-encoded sequential semantic information, the SSF component demonstrates a clear performance advantage, leading to significant improvements on both datasets.

### 4.3.4 Effectiveness of BHGAT component

To comprehensively assess the superiority of this component, we conduct experiments on two datasets: SemEval 2016 Task 6A and ABAM ar-

gument segment datasets. In our experiments, we incorporate the BHGAT component into the BERT-based framework and compare the experimental results, as shown in Table 7.

Table 7 clearly demonstrate that the inclusion of the BHGAT component has resulted in significant performance improvements. This improvement can be attributed to several key factors. Firstly, the BHGAT component has the ability to capture and leverage information from multiple samples that share the same topic. By considering the expressions of different stances and the embedding representations of words in different contexts, the BHGAT component enhances the model's discriminative power and facilitates more accurate stance detection. Furthermore, the versatility of the BHGAT component is noteworthy. It can be seamlessly integrated into various frameworks, enabling performance enhancements across different models. This flexibility makes the BHGAT component highly adaptable, particularly in classification tasks that involve topic information, such as stance detection.

### 4.3.5 Effectiveness of SMIA component

The SMIA component introduced in the ATE module aims to restrict the range of aspect term recognition. To assess its effectiveness, we present the confusion matrix based on the Token-Nested evaluation index in figure 3.

In figure 3, the confusion matrix is presented with a dimension of 6x6. It is important to note that in real data, there is no combination of the aspect label **ASP** and the stance label **NON**, as aspect terms only exist within argument units. As a result, the

Table 6: Performance comparison of results in ABAM and AURC-8.

| | ABAM | | | | | | | AURC-8 | | | |
| | Segment-Level | | | | | Token-Level | | Segment-Level | | | |
| | ASP | CON | PRO | MicF1 | MacF1 | Token-Flat | Token-Nested | CON | PRO | MicF1 | MacF1 |
|---|---|---|---|---|---|---|---|---|---|---|---|
| LSTM-CRF | 0.7456 | 0.3890 | 0.3879 | 0.6434 | 0.5075 | 0.7633 | 0.5373 | 0.3264 | 0.3350 | 0.3308 | 0.3307 |
| SSF-CRF | **0.7473** | **0.3995** | **0.3995** | **0.6485** | **0.5154** | **0.7762** | **0.5526** | **0.3329** | **0.3585** | **0.3458** | **0.3457** |

Table 7: Performance comparison of results in SemEval 2016 task 6A and ABAM argument segment.

| | SemEval 2016 task 6A | | | | | ABAM argument segment | | | |
| | CON | NONE | PRO | MicF1 | MacF1 | CON | PRO | MicF1 | MacF1 |
|---|---|---|---|---|---|---|---|---|---|
| BERT | **0.7551** | 0.5885 | 0.6436 | 0.6942 | 0.6624 | 0.7989 | 0.7881 | 0.7937 | 0.7935 |
| BERT-BHGAT | 0.7541 | **0.6071** | **0.6591** | **0.6990** | **0.6734** | **0.8126** | **0.7908** | **0.8023** | **0.8017** |

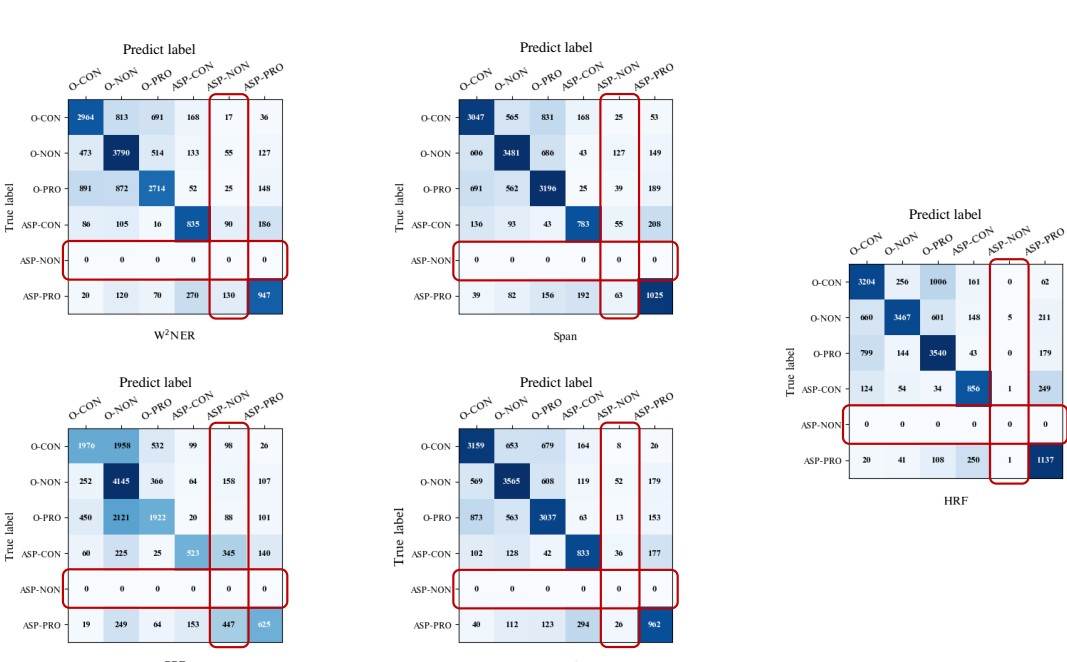

Figure 3: The confusion matrices for different models

fifth row of the confusion matrix is always zero. In the confusion matrix, we focus on the fifth column of each confusion matrix. The model identifies aspect terms and argument units separately, during the prediction process, if the prediction range of term is not constrained, it may generate a wrong match between the aspect term label ASP and the stance label NON. However, by observing the fifth column of each confusion matrix, we can observe a significant reduction in misjudgment after adding the span mask constraints imposed by the SMIA component. This outcome reinforces the effectiveness of the SMIA component in constraining the recognition of aspect terms.

## 5 Conclusion

This paper presents a novel layer-based approach for the aspect-based argument mining task, utilizing a hierarchical enhancement framework consisting of four modules: basic module, argument unit enhancement module, aspect term enhancement module, and decision module. The SSF component plays a crucial role in optimizing underlying representations, which can be utilized across various tasks. It enhances the framework's capability by incorporating syntactic information into the encoding process, improving performance on sequence labeling tasks. The BHGAT component, effective for classification tasks involving topic information, enhances the framework's generalization capabilities. The SMIA component is specifically designed for aspect-based argument mining tasks, aiming to constrain the recognition range of aspect terms. It effectively improves the accuracy of aspect term recognition and contributes to the overall performance of the framework.

## Limitations

However, it should be noted that the proposed BH-GAT is currently only suitable for classification tasks with topic information. Its generalization to more general tasks needs further investigation in our future work. In addition, our current framework has primarily focused on adopting a layer-based method for Nested Named Entity Recognition (NNER), without extensively exploring how to mine the correlation between argument units and aspect terms. In future work, it is essential to delve deeper into the correlation between these two entities and fully utilize the guiding information between them.

## Acknowledgements

The authors would like to thank all anonymous reviewers for their valuable comments and suggestions which have significantly improved the quality and presentation of this paper. The works described in this paper are supported by the National Key Research and Development Program of China (2022QY0300-01), National Natural Science Foundation of China (62106130, 62076158, 62072294, 62272286), Natural Science Foundation of Shanxi Province, China (20210302124084), Scientific and Technological Innovation Programs of Higher Education Institutions in Shanxi, China (2021L284), and CCF-Zhipu AI Large Model Foundation of China (CCF-Zhipu202310).

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
