# OpenReview forum: "Hierarchical Enhancement Framework for Aspect-based Argument Mining"
_EMNLP/2023/Conference — EMNLP 2023 Findings_

### Official Review · Reviewer_qcN8 · 2023-07-27

**Soundness:** 3

**Excitement:**

3: Ambivalent: It has merits (e.g., it reports state-of-the-art results, the idea is nice), but there are key weaknesses (e.g., it describes incremental work), and it can significantly benefit from another round of revision. However, I won't object to accepting it if my co-reviewers champion it.

**Paper Topic And Main Contributions:**

The paper proposes a layer-based Hierarchical Enhancement Framework for the ABAM task to efficiently overlook the need for tailored strategies to effectively address the specific challenges of ABAM tasks, and introduces three novel components: the Semantic and Syntactic Fusion component, the Batch-level Heterogeneous Graph Attention Network component, and the Span Mask Interactive Attention component. The approach is evaluated effective on multiple datasets and multiple tasks, with further ablation study.

**Reasons To Accept:**

1、The paper propused novel module the BHGAT component.

**Reasons To Reject:**

1、Semantic and Syntactic Fusion component proposed in the paper are not innovative. The author's explanation of why this component is used lacks clarity.

2、The paper does not consider many state-of-the-art ABAM methods for comparison, it has been noted that the compared baseline models do not adequately address the ABAM task but rather other tasks. Therefore, the comparison between the proposed method and the baseline model fails to convincingly demonstrate its advancement.

3、Equations and model structure diagrams, being essential components of your research, should align and corroborate with each other. However, in this paper, there are evident discrepancies, such as eq9, eq10, eq11.

4、In the second part of the narrative, some of the elaborations are not clear enough, for example， in the Basic Module, the meaning of the symbols W and U and Q is not given.

5、Lack of clarity in expression: The writing of the article could be improved, it lacks clarity and precision, and certain paragraphs are too colloquial.

**Reproducibility:**

4: Could mostly reproduce the results, but there may be some variation because of sample variance or minor variations in their interpretation of the protocol or method.

**Reviewer Confidence:**

4: Quite sure. I tried to check the important points carefully. It's unlikely, though conceivable, that I missed something that should affect my ratings.

---

> ### Author Rebuttal · Authors · 2023-08-27
>
> QA1: Semantic and Syntactic Fusion component proposed in the paper are not innovative. The author's explanation of why this component is used lacks clarity.
>
>     Indeed, your observation is accurate. The primary contribution of this article rests with the BHGAT component, closely followed by SMIA. BHGAT introduces a novel functionality wherein it facilitates the exchange of opinion expressions across distinct argument units centered around aspect terms within the same topic and batch. On the other hand, SMIA, tailored specifically for ABAM tasks, adeptly confines the scope of aspect term identification.
>
>     Additionally, the SSF component, drawing inspiration from existing research, generates components suited for singular presentations. These components serve the purpose of fine-tuning downstream tasks with BERT. While a corresponding reference for this aspect will be included subsequently, it's noteworthy that this won't be presented as an innovative point within the contribution section of the introduction. In our revised paper, we will highlight the other two main contributions, namely BHGAT component and SMIA component, ensuring the clarity and accuracy of the article's contributions.
>
>
> QA2: The paper does not consider many state-of-the-art ABAM methods for comparison, it has been noted that the compared baseline models do not adequately address the ABAM task but rather other tasks. Therefore, the comparison between the proposed method and the baseline model fails to convincingly demonstrate its advancement.
>
>     ABAM, initially introduced by Trautmann at the 2020 Argument Mining workshop of ACL, has not garnered sufficient subsequent research outputs. Our current study aims to bridge this gap by harnessing five recent baselines from the years 2021 and 2022. While we acknowledge the significance of expanding our comparative spectrum, we concur with your viewpoint on the necessity of a more comprehensive evaluation strategy. In direct response to your inquiry, we have augmented our baseline repertoire with an additional model grounded in contemporary advancements – the ChatGLM-6B, an open-source and lightweight alternative to ChatGPT. It functions as an advanced language processing model imbued with extensive capabilities. The ChatGLM-6B has the following distinctive characteristics:
>
>     Has about 6.2 billion parameters.
>     The model is pre-trained on 1 trillion tokens—equally from English and Chinese.
>     Subsequently, techniques such as supervised fine-tuning and reinforcement learning with human feedback are used.
>
>     Next, we outline how ChatGLM-6B can be effectively applied to the ABAM task. We construct a prompt structured as follows: “{“content”: topic is “xxx”, text is “xxxxx”, “summary”: “CON Argument Unit”--->“xxxxx”, “PRO Argument Unit”--->“xxxxx”, aspect term--->“xxxxx”}”. The model undergoes fine-tuning using the ABAM dataset and is subsequently evaluated on the test dataset to yield final experimental results.
>
>     Our findings once again verify the superiority of our proposed HEF framework when compared with the ChatGLM-6B, both for the segment-level and token-level analyses. These outcomes not only reinforce the efficacy of our approach but also underline its significance in the domain. We trust that these results will effectively allay any concerns you might have had.
>
>     Building upon your insightful suggestion, we are dedicated to extending our experimental endeavors. Our forthcoming plans encompass the incorporation of additional comparative methodologies. We are committed to meticulously documenting and incorporating the outcomes of these novel experiments into the final paper, thereby enriching the overall robustness of our research.
>
>                        Table 1 Comparison with large language model
>     --------------------------------------------------------------------------------------
>     |	   |                  Segment-Level              |          Token-Level      |
>     |          |   ASP   |  CON   |  PRO   |  MicF1 | MacF1  | Token-Flat | Token-Nested |
>     |ChatGLM-6B|  0.7010 | 0.4402 | 0.4311 | 0.6321 | 0.5241 |  0.7522    |    0.5981    |
>     |HEF(our)  |  0.7969 | 0.6283 | 0.6176 | 0.7494 | 0.6810 |  0.8327    |   0.6204     |
>     --------------------------------------------------------------------------------------
>
> QA3: Equations and model structure diagrams, being essential components of your research, should align and corroborate with each other. However, in this paper, there are evident discrepancies, such as eq9, eq10, eq11.
>
>     Thank you for your valuable feedback. We have taken note of the confusion arising from the representation of the gated unit's connection with "$g_t$" in Figure 2, and we have since rectified this issue. Your input has been instrumental in improving the clarity and accuracy of our depiction.
>
> QA4: In the second part of the narrative, some of the elaborations are not clear enough, for example， in the Basic Module, the meaning of the symbols W and U and Q is not given.
>
>     We deeply appreciate your guidance, and we are committed to incorporating the recommended enhancements. We intend to provide further elucidation for the formula in question. Specifically, we will offer clarifications for the parameter matrices W, U, and Q. These matrices correspond respectively to the semantic representation $x_t$, the preceding state $h_{t-1}$, and the syntactic representation of $g_t$. Your input has proved invaluable in our pursuit of rendering the material more comprehensible and transparent.
>
> QA5: Lack of clarity in expression: The writing of the article could be improved, it lacks clarity and precision, and certain paragraphs are too colloquial.
>
>     We recognize the need to enhance the article's readability and clarity by incorporating additional explanatory details. Concurrently, we're dedicated to refining the article's focus by streamlining and eliminating extraneous descriptions. We'll proceed to adjust the writing accordingly in alignment with these objectives.
>
>     We extend our sincere gratitude for your valuable feedback and perceptive comments, which play a pivotal role in elevating the caliber of this paper. Your insights have proven indispensable in our continuous efforts to enhance its quality.

---

### Official Review · Reviewer_3LjC · 2023-08-02

**Typos Grammar Style And Presentation Improvements:** 1.In line 57, 61, two errors arise in…
**Soundness:** 3

**Excitement:**

3: Ambivalent: It has merits (e.g., it reports state-of-the-art results, the idea is nice), but there are key weaknesses (e.g., it describes incremental work), and it can significantly benefit from another round of revision. However, I won't object to accepting it if my co-reviewers champion it.

**Missing References:**

[1]	Yuanhe Tian, Guimin Chen, Yan Song, and Xiang Wan. 2021. Dependency-driven Relation Extraction with Attentive Graph Convolutional Networks. In Proceedings of the 59th Annual Meeting of the Association for Computational Linguistics and the 11th International Joint Conference on Natural Language Processing (Volume 1: Long Papers), pages 4458–4471.
[2]	Li, Y., Li, Z., Zhang, M., Wang, R., Li, S., & Si, L. (2019). Self-attentive Biaffine Dependency Parsing. In IJCAI (pp. 5067-5073).

**Paper Topic And Main Contributions:**

This paper focuses on the Aspect-Based Argument Mining (ABAM) task and proposes a novel Hierarchical Enhancement Framework (HEF) consisting of four modules to solve three key challenges of ABAM. The experimental results demonstrate the effectiveness of the proposed method.

**Questions For The Authors:**

Question A: In Basic Module, obtaining the two representations uses BERT and LSTM, respectively. Is there anything special about this?
Question B: Compared to the other methods, what are HEF's strengths in addressing the three key challenges in ABAM?
Question C: In the experiment, how does HEF perform when all the topics are mixed together?


**Reasons To Accept:**

1. The studied problem is interesting and important. In general, this paper is well-written and organized.
2. The experimental results show that the proposed method significantly outperforms state-of-the-art baselines


**Reasons To Reject:**

1.The Semantic and Syntactic Fusion (SSF) component aiming to bridge the gap between distant words is not novel. The topic of long-distance dependencies between words has been studied in previous works [1][2]. It’s not clear how the proposed method advances from them.
2.The overall framework figure is not clear. This figure does not illustrate The key component modules (Figure 2).
3. This paper does not clearly explain how the three key challenges were addressed, and each module does not correspond to a challenge.

[1]	Yuanhe Tian, Guimin Chen, Yan Song, and Xiang Wan. 2021. Dependency-driven Relation Extraction with Attentive Graph Convolutional Networks. In Proceedings of the 59th Annual Meeting of the Association for Computational Linguistics and the 11th International Joint Conference on Natural Language Processing (Volume 1: Long Papers), pages 4458–4471.
[2]	Li, Y., Li, Z., Zhang, M., Wang, R., Li, S., & Si, L. (2019). Self-attentive Biaffine Dependency Parsing. In IJCAI (pp. 5067-5073).


**Reproducibility:**

4: Could mostly reproduce the results, but there may be some variation because of sample variance or minor variations in their interpretation of the protocol or method.

**Reviewer Confidence:**

4: Quite sure. I tried to check the important points carefully. It's unlikely, though conceivable, that I missed something that should affect my ratings.

---

> ### Author Rebuttal · Authors · 2023-08-27
>
> QA1: The Semantic and Syntactic Fusion (SSF) component aiming to bridge the gap between distant words is not novel. The topic of long-distance dependencies between words has been studied in previous works [1][2]. It’s not clear how the proposed method advances from them.
>
>     Yes, you are right that the main contribution of this article is the BHGAT component, followed by SMIA. BHGAT is a new component that can communicate the expression of opinions between different argument units around aspect terms under the same topic in the same batch. SMIA is an ABAM task-specific component that effectively constrains the identification scope of aspect terms. Drawing on other work, SSF generates components for just one presentation that can be used to fine-tune downstream tasks with BERT. The corresponding reference will be added later. And it will be removed in the contribution part of introduction and not put forward as an innovative point.
>
>     Indeed, you are correct in identifying the primary contribution of this article, which lies prominently within the BHGAT component, followed by the SMIA module. The BHGAT introduces a novel capability to facilitate the exchange of opinion expressions across different argument units centered around aspect terms within the same topic and batch. Meanwhile, the SMIA component caters specifically to ABAM tasks, skillfully limiting the scope for aspect term identification.
>
>     Expanding on this, the SSF component, derived from prior work, generates elements optimized for fine-tuning BERT with respect to individual presentations. Although a corresponding reference will be incorporated later, it's important to highlight that this aspect will not be presented as an innovative point in the contribution section of the introduction, and will subsequently be removed from there. In our revised paper, we will highlight the other two main contributions, namely BHGAT component and SMIA component, ensuring the clarity and accuracy of the article's contributions.
>
> QA2: The overall framework figure is not clear. This figure does not illustrate The key component modules (Figure 2).
>
>     Figure 2 illustrates our endeavor to visually depict the three pivotal component modules, each occupying a prominent section of the figure. Following your suggestion, our intention for the final version is to enhance their distinction by employing distinct colors for each module, to further highlight them clearly.
>
> QA3: This paper does not clearly explain how the three key challenges were addressed, and each module does not correspond to a challenge.
>
>     In the introductory section, we've delineated three distinct challenges:
>     1. Constructing a robust foundational representation capable of effectively encapsulating contextual information.
>     2. Unearthing the correlations between opinion expressions associated with various stances within the context of the same topic.
>     3. Harnessing task-specific attributes to enhance the extraction of argument units and aspect terms.
>
>     Our three modules are meticulously tailored to directly address these challenges:
>     1. SSF Module: This module is devoted to constructing a robust foundational representation, ensuring the effective encoding of contextual nuances.
>     2. BHGAT Module: Our BHGAT components are strategically designed to uncover and exploit correlations among opinion expressions that align with different stances under a shared topic.
>     3. SMIA Module: Focused on task-specific enhancements, the SMIA components serve to bolster the extraction of argument units and aspect terms.
>
>     In essence, each of our three modules takes a dedicated approach to tackling one of these challenges, effectively aligning with them in a one-to-one manner. We will make it clear in our final version.
>
> Questions For The Authors:
>
> Question A: In Basic Module, obtaining the two representations uses BERT and LSTM, respectively. Is there anything special about this?
>
>     BERT, as a general language model, often requires fine-tuning in downstream tasks. BERT and LSTM are common combined operations. The same operation is used in the following papers. We will give corresponding quotes later in the article.
>
>     BERT, functioning as a broad-spectrum language model, frequently necessitates fine-tuning when employed in downstream tasks. The combination of BERT and LSTM is a prevalent practice, consistently applied across various works within the field. This approach is mirrored in the subsequent papers, and we will provide relevant citations to substantiate this assertion later in our article.
>
>     [1] Jingye Li, Hao Fei, Jiang Liu, Shengqiong Wu, Meishan Zhang, Chong Teng, Donghong Ji, and Fei Li. 2022. Unified named entity recognition as word-word relation classification. In Proceedings of the AAAI Conference on Artificial Intelligence, volume 36, pages 10965–10973.
>     [2] He Zhao, Longtao Huang, Rong Zhang, Quan Lu, and Hui Xue. Spanmlt: A span-based multi-task learning framework for pair-wise aspect and opinion terms extraction. Proceedings of the 58th Annual Meeting of the Association for Computational Linguistics. 2020: 3239-3248.
>
>     The combination of BERT and LSTM is a prevalent practice due to the complementary strengths these two models offer. BERT excels at capturing intricate contextual information and semantic nuances, while LSTM is adept at modeling sequential dependencies and patterns within data. Integrating the two models allows for a synergistic approach where BERT's contextual understanding is enriched by LSTM's sequential processing capabilities. This combination often leads to improved performance, particularly in tasks that involve both contextual understanding and sequential reasoning. BERT's pre-trained representations provide a solid foundation, which LSTM can then build upon by capturing temporal dependencies and context coherence. This joint utilization taps into the best of both worlds, resulting in enhanced results across a range of natural language processing tasks.
>
> Question B: Compared to the other methods, what are HEF's strengths in addressing the three key challenges in ABAM?
>
>     In the ablation experiment (as depicted in Table 5), we delve into the impact of incrementally incorporating components on the experimental outcomes. Notably, the SSF components play a pivotal role in crafting resilient model representations that offer tangible benefits for both argument unit identification and attribute item extraction. On the other hand, the BHGAT module stands out in its capacity to proficiently convey diverse argument unit expressions under a common topic, thereby lending crucial support to the argument unit identification process.
>
>     Furthermore, the SMIA component assumes a distinct role by refining the scope within which attribute items are identified. This strategic refinement, in turn, yields a marked enhancement in the efficacy of attribute item extraction. This comprehensive combination of modules illustrates their interdependent contributions towards refining and amplifying the overall performance of the model.
>
> Question C: In the experiment, how does HEF perform when all the topics are mixed together?
>
>     During the experiment, we have amalgamated all topics to conduct a comprehensive analysis. Through the integration of each individual component, we optimized their respective performances. Additionally, the utilization of topic information further enhanced the overall model performance. The precise outcomes of these manipulations are elaborated upon in the ablation experiment results, presented in Table 5.
>
> Typos Grammar Style And Presentation Improvements:
>
> 1.In line 57, 61, two errors arise in using the English a, an (i.e., an argument unit). 2.In line 174, the sentence “The characters representation within $w^{text}_i$ as $w^{char}_i$.” is unclear.
>
>     We greatly appreciate your valuable suggestions and insightful comments. As a result, we have carefully reviewed the manuscript and rectified any identified errors. Your input has contributed significantly to the refinement of our work.

---

### Official Review · Reviewer_1g4G · 2023-08-05

**Paper Topic And Main Contributions:** 1. This paper proposes a general SSF …
**Soundness:** 3

**Excitement:**

3: Ambivalent: It has merits (e.g., it reports state-of-the-art results, the idea is nice), but there are key weaknesses (e.g., it describes incremental work), and it can significantly benefit from another round of revision. However, I won't object to accepting it if my co-reviewers champion it.

**Questions For The Authors:**

The same to the above the section "Reasons To Reject".

**Reasons To Accept:**

1. The idea of hierarchical enhancement framework for aspect-based argument mining is innovative and intuitive.
2. This paper clearly and logically describes its contribution to hierarchical enhancement framework for aspect-based argument mining. From the extensive and comprehensive experimental results of experiments in this paper, the proposed framework demonstrates significant performance improvements over existing approaches..
3. The framework studied in this paper is interesting and well motivated and the paper is well written.
4. Experiments are tested on real datasets.

**Reasons To Reject:**

1. It is better to choose more baselines from different articles for comparison.
2. The paper lacks related work.
3. Experiments on the effectiveness of SMIA component could be added to make the experiments more comprehensive.
4. Compared with LSTM-CRF, although the performance of SF-CRF is slightly better, will the computational complexity increase dramatically?

**Reproducibility:**

3: Could reproduce the results with some difficulty. The settings of parameters are underspecified or subjectively determined; the training/evaluation data are not widely available.

**Reviewer Confidence:**

4: Quite sure. I tried to check the important points carefully. It's unlikely, though conceivable, that I missed something that should affect my ratings.

---

> ### Author Rebuttal · Authors · 2023-08-27
>
> QA1: It is better to choose more baselines from different articles for comparison.
>
>     ABAM, initially introduced by Trautmann at the 2020 Argument Mining workshop of ACL, has yet to yield substantial subsequent research contributions. To address this gap, our present study aims to rectify the situation by leveraging five recent baselines from the years 2021 and 2022. While we recognize the importance of broadening our comparative scope, we share your perspective on the need for a more comprehensive evaluation strategy. In direct response to your query, we have expanded our baseline repertoire with an additional model rooted in contemporary advancements – the ChatGLM-6B. This open-source and lightweight alternative to ChatGPT is an advanced language processing model imbued with extensive capabilities. Noteworthy attributes of ChatGLM-6B include:
>
>     Has about 6.2 billion parameters.
>     The model is pre-trained on 1 trillion tokens—equally from English and Chinese.
>     Subsequently, techniques such as supervised fine-tuning and reinforcement learning with human feedback are used.
>
>     Next, we outline how ChatGLM-6B can be effectively applied to the ABAM task. We construct a prompt structured as follows: “{“content”: topic is “xxx”, text is “xxxxx”, “summary”: “CON Argument Unit”--->“xxxxx”, “PRO Argument Unit”--->“xxxxx”, aspect term--->“xxxxx”}”. The model undergoes fine-tuning using the ABAM dataset and is subsequently evaluated on the test dataset to yield final experimental results.
>
>     Our findings once again verify the superiority of our proposed HEF framework when compared with the ChatGLM-6B, both for the segment-level and token-level analyses. These outcomes not only reinforce the efficacy of our approach but also underline its significance in the domain. We trust that these results will effectively allay any concerns you might have had.
>
>     Building upon your insightful suggestion, we are dedicated to extending our experimental endeavors. Our forthcoming plans encompass the incorporation of additional comparative methodologies. We are committed to meticulously documenting and incorporating the outcomes of these novel experiments into the final paper, thereby enriching the overall robustness of our research.
>
>                        Table 1 Comparison with large language model
>     --------------------------------------------------------------------------------------
>     |	   |                  Segment-Level              |          Token-Level      |
>     |          |   ASP   |  CON   |  PRO   |  MicF1 | MacF1  | Token-Flat | Token-Nested |
>     |ChatGLM-6B|  0.7010 | 0.4402 | 0.4311 | 0.6321 | 0.5241 |  0.7522    |    0.5981    |
>     |HEF(our)  |  0.7969 | 0.6283 | 0.6176 | 0.7494 | 0.6810 |  0.8327    |   0.6204     |
>     --------------------------------------------------------------------------------------
>
> QA2: The paper lacks related work.
>
>     In our initial submission, owing to constraints on available space, we have amalgamated the discussion on Related work with the Introduction section. We will include an independent Related work Session in the final version of our paper to include more relevant research papers. Please refer below for a more comprehensive overview of our Related Work section. We are open to incorporating further references should you have any suggestions on additional pertinent research. Your input is highly appreciated.
>
>     Related Work
>
>     Below, we will present an overview of two relevant areas of research, namely Aspect-based Argument Mining and Named Entity Recognition.
>
>     Aspect-based Argument Mining:
>
>     The objective of Misra et al. [1] is to identify specific arguments and counter-arguments in social media texts,  categorize them into different aspects, and utilize this aspect information to generate argument summaries. Similarly, Misra et al. [2] focus on inducing and identifying argument aspects across multiple conversations, ranking the extracted arguments based on their similarity, and generating corresponding summaries. However, these earlier works have been limited to a few specific topics. In recent research, the approach has been extended to cover a broader range of 28 topics, introducing a novel corpus for aspect-based argument clustering [3]. Furthermore, Gemechu and Reed [4] decompose propositions into four functional components: aspects, target concepts, and opinions on aspects and target concepts. By leveraging the relationships among these components, they infer argument relations and gain a deeper understanding of the argument structure. In a different study, Bar-Haim et al. [5] focus on summarizing the arguments, supporting each side of a debate, mapping them to a concise list of key points, which are similar to the aspect terms highlighted earlier. Lastly, Trautmann [6] redefines the aspect-based argument mining task based on clause-level argument unit recognition and classification in heterogeneous document collections [7].
>
>     Named Entity Recognition:
>
>     The related work on Named Entity Recognition (NER) methods can be broadly classified into two categories: token-based NER and span-based NER.
>
>     Token-based NER is typically framed as a sequence labeling task, where each word is assigned a label indicating its entity type. The widely used approach for token-based NER involves utilizing a bidirectional LSTM as an encoder to learn contextual representations of words, coupled with Conditional Random Fields (CRFs) [8] as a decoder to label the words [9]. Some researchers have also incorporated character-level representations to capture spelling features. These character-level and word-level embeddings are concatenated and fed as input to a BiLSTM with CRF for labeling [10, 11]. More recently, the field of NER has experienced notable progress, with the introduction of powerful pre-trained language models, such as ELMo and BERT [12, 13]. These models have been widely adopted to enhance NER performance.
>
>     Span-based NER approaches, on the other hand, involve enumerating all possible spans in sentences and classifying each span into a specific entity category (e.g., PER, ORG, LOC). For example, Li et al. [14] view NER as a Machine Reading Comprehension (MRC) task, where entities are extracted as answer spans. Yu et al. [15] propose an approach that extracts spans based on start and end positions, and employ biaffine attention to measure the likelihood of a text span being a mention. Furthermore, Shen et al. [16] present a two-stage NER model that generates span proposals first and then classifies them into corresponding entity categories. While span-based models offer a fine-grained approach to NER, they may encounter certain limitations related to the maximum span length and computational complexity inherent to their enumerative nature. Particularly, enumerating all possible spans can lead to a significant increase in the number of candidate spans, making the task computationally expensive.
>
>     [1] Amita Misra, Pranav Anand, Jean E. Fox Tree, and Marilyn Walker. Using summarization to discover argument facets in online idealogical dialog. In Proceedings of the 2015 Conference of the North American Chapter of the Association for Computational Linguistics: Human Language Technologies, pages 430–440, 2015.
>     [2] Amita Misra, Brian Ecker, and Marilyn Walker. Measuring the similarity of sentential arguments in dialogue. In Proceedings of the 17th Annual Meeting of the Special Interest Group on Discourse and Dialogue, pages 276–287, 2016.
>     [3] Nils Reimers, Benjamin Schiller, Tilman Beck, Johannes Daxenberger, Christian Stab, and Iryna Gurevych. Classification and clustering of arguments with contextualized word embeddings. In Proceedings of the 57th Annual Meeting of the Association for Computational Linguistics, pages 567–578, 2019.
>     [4] Debela Gemechu and Chris Reed. Decompositional argument mining: A general purpose approach for argument graph construction. In Proceedings of the 57th Annual Meeting of the Association for Computational Linguistics, pages 516–526, 2019.
>     [5] Roy Bar-Haim, Lilach Eden, Roni Friedman, Yoav Kantor, Dan Lahav, and Noam Slonim. From arguments to key points: Towards automatic argument summarization. In Proceedings of the 58th Annual Meeting of the Association for Computational Linguistics, pages 4029–4039, 2020.
>     [6] Dietrich Trautmann. Aspect-based argument mining. In Proceedings of the 7th Workshop on Argument Mining, pages 41–52, 2020.
>     [7] Dietrich Trautmann, Johannes Daxenberger, Christian Stab, Hinrich Schütze, and Iryna Gurevych. Fine-grained argument unit recognition and classification. In Proceedings of the AAAI Conference on Artificial Intelligence, pages 9048–9056, 2020.
>     [8] John D. Lafferty, Andrew McCallum, and Fernando C. N. Pereira. Conditional random fields: Probabilistic models for segmenting and labeling sequence data. In Proceedings of the Eighteenth International Conference on Machine Learning, page 282–289, 2001.
>     [9] Rrubaa Panchendrarajan and Aravindh Amaresan. Bidirectional LSTM-CRF for named entity recognition. In Proceedings of the 32nd Pacific Asia Conference on Language, Information and Computation, pages 531–540, 2018.
>     [10] Fei Li, Zheng Wang, Siu Cheung Hui, Lejian Liao, Dandan Song, Jing Xu, Guoxiu He, and Meihuizi Jia. Modularized interaction network for named entity recognition. In Proceedings of the 59th Annual Meeting of the Association for Computational Linguistics and the 11th International Joint Conference on Natural Language Processing, pages 200–209, 2021.
>     [11] Ying Luo, Fengshun Xiao, and Hai Zhao. Hierarchical contextualized representation for named entity recognition. In Proceedings of the Thirty-Fourth AAAI Conference on Artificial Intelligence, pages 8441–8448, 2020.
>     [12] Jacob Devlin, Ming-Wei Chang, Kenton Lee, and Kristina Toutanova. BERT: Pre-training of deep bidirectional transformers for language understanding. In Proceedings of the 2019 Conference of the North American Chapter of the Association for Computational Linguistics: Human Language Technologies, Volume 1 (Long and Short Papers), pages 4171–4186, 2019.
>     [13] Matthew E. Peters, Mark Neumann, Mohit Iyyer, Matt Gardner, Christopher Clark, Kenton Lee, and Luke Zettlemoyer. Deep contextualized word representations. In Proceedings of the 2018 Conference of the North American Chapter of the Association for Computational Linguistics: Human Language Technologies, pages 2227–2237, 2018.
>     [14] Xiaoya Li, Jingrong Feng, Yuxian Meng, Qinghong Han, Fei Wu, and Jiwei Li. A unified MRC framework for named entity recognition. In Proceedings of the 58th Annual Meeting of the Association for Computational Linguistics, pages 5849–5859, 2020.
>     [15] Juntao Yu, Bernd Bohnet, and Massimo Poesio. Named entity recognition as dependency parsing. In Proceedings of the 58th Annual Meeting of the Association for Computational Linguistics, pages 6470–6476, 2020.
>     [16] Yongliang Shen, Xinyin Ma, Zeqi Tan, Shuai Zhang, Wen Wang, and Weiming Lu. Locate and label: A two-stage identifier for nested named entity recognition. In Proceedings of the 59th Annual Meeting of the Association for Computational Linguistics and the 11th International Joint Conference on Natural Language Processing, pages 2782–2794, 2021.
>
>
> QA3: Experiments on the effectiveness of SMIA component could be added to make the experiments more comprehensive.
>
>     We fully agree with you. Following your suggestion, we have demonstrated the Superiority of SMIA in Label Combination through Confusion Matrix Analysis. In particular, to validate the efficacy of the SMIA component in terms of label combination, we have conducted a comprehensive analysis of the confusion matrix. We intend to include the following experimental aspect in our paper.
>
>     Evaluating the Efficacy of SMIA in the ATE Module:
>
>     Our analysis involves a 6x6 dimensional confusion matrix that encompasses the following label combinations: [NON, O], [NON, ASP], [PRO, O], [PRO, ASP], [CON, O], and [CON, ASP]. It's worth noting that, in real-world data, the combination of the aspect label ASP and the stance label NON is non-existent, as aspect terms exclusively manifest within argument units. Consequently, the fifth row of the confusion matrix consistently remains at zero.
>
>     Our particular focus lies in the fifth column of each confusion matrix. The model distinguishes aspect terms and argument units as separate entities during the prediction process. Without constraints on the prediction range of terms, there is a potential for incorrect matches to emerge between the aspect term label ASP and the stance label NON. However, upon scrutinizing the fifth column of each confusion matrix, a noticeable reduction in misjudgments becomes evident subsequent to the implementation of span mask constraints imposed by the SMIA component.
>
>     This empirical evidence strongly bolsters the notion of the SMIA component's effectiveness in precisely constraining the recognition of aspect terms, reaffirming its significant role in refining the ATE module.
>
>
>
> QA4: Compared with LSTM-CRF, although the performance of SF-CRF is slightly better, will the computational complexity increase dramatically?
>
>
>     The SSF component serves the purpose of refining BERT representations to align with downstream tasks, thereby establishing a robust foundational representation that enhances overall model performance. Our aim is to elucidate the temporal aspects associated with training LSTM-CRF and SF-CRF over the span of one epoch using two distinct datasets.
>
>     You are absolutely right that the incorporation of SSF introduces heightened computational complexity, attributed to the inclusion of graph attention neural networks and supplementary gating mechanisms. Consequently, this addition leads to an increase in time-related costs, as shown in Table 2.
>
>     However, within the context of ABAM tasks, the trade-off of investing additional time is justified by the resultant enhancement in model performance. The heightened computational demands brought about by SSF align with the objective of refining the model's capabilities in addressing the intricacies of ABAM tasks, substantiating the rationale for embracing this time-cost exchange. In addition, as computational power increases, the impact of extended processing times becomes less significant.
>
>
>        Table 2 Time cost for LSTM-CRF and SSF-CRF to train an epoch
>                 -------------------------------------------
>                 |Model         |  Dataset |   Time/epoch  |
>                 |LSTM-CRF      |   AURC   |    53.631s    |
>                 |              |   ABAM   |    54.692s    |
>                 |SSF-CRF       |   AURC   |    119.061s   |
>                 |              |   ABAM   |    127.293s   |
>                 -------------------------------------------

---

### Meta-Review · Area_Chair_wNcM · 2023-09-19

**Recommendation:** 3

**Metareview:**

This paper addresses the Aspect-Based Argument Mining (ABAM) task and proposes a Hierarchical Enhancement Framework (HEF) consisting of four modules for ABAM. This paper studies an important problem and it is well written. Experimental results show that the proposed model outperforms baseline models. The paper needs to add a relation work section. Experiments on the effectiveness of SMIA component could be added as well. This paper should also clearly explain how the three key challenges were addressed by the proposed components.

---

### Decision · Program_Chairs · 2023-10-07

**Decision:**

Accept-Findings

**Comment:**

This paper addresses the Aspect-Based Argument Mining (ABAM) task and proposes a Hierarchical Enhancement Framework (HEF) consisting of four modules for ABAM. This paper studies an important problem and it is well written. Experimental results show that the proposed model outperforms baseline models. The paper needs to add a relation work section. Experiments on the effectiveness of SMIA component could be added as well. This paper should also clearly explain how the three key challenges were addressed by the proposed components.